# High Fidelity Neural Audio Compression

**Alexandre Défossez**[*]  *defossez@meta.com*
*Meta AI, FAIR Team, Paris, France*

**Jade Copet**[*]  *jadecopet@meta.com*
*Meta AI, FAIR Team, Paris, France*

**Gabriel Synnaeve**[†]  *gab@meta.com*
*Meta AI, FAIR Team, Paris, France*

**Yossi Adi**[†]  *adiyoss@meta.com*
*Meta AI, FAIR Team, Tel-Aviv, Israel*

**Reviewed on OpenReview:** *https://openreview.net/forum?id=ivCd8z8zR2*

## Abstract

We introduce a state-of-the-art real-time, high-fidelity, audio codec leveraging neural networks. It consists in a streaming encoder-decoder architecture with quantized latent space trained in an end-to-end fashion. We simplify and speed-up the training by using a single multiscale spectrogram adversary that efficiently reduces artifacts and produce high-quality samples. We introduce a novel loss balancer mechanism to stabilize training: the *weight* of a loss now defines the fraction of the overall gradient it should represent, thus decoupling the choice of this hyper-parameter from the typical scale of the loss. Finally, we study how lightweight Transformer models can be used to further compress the obtained representation by up to 40%, while staying faster than real time. We provide a detailed description of the key design choices of the proposed model including: training objective, architectural changes and a study of various perceptual loss functions. We present an extensive subjective evaluation (MUSHRA tests) together with an ablation study for a range of bandwidths and audio domains, including speech, noisy-reverberant speech, and music. Our approach is superior to the baselines methods across all evaluated settings, considering both 24 kHz monophonic and 48 kHz stereophonic audio. Code and samples are available under github.com/facebookresearch/encodec.

## 1 Introduction

Recent studies suggest that streaming audio and video have accounted for the majority of the internet traffic in 2021 (82% according to (Cisco, 2021)). With the internet traffic expected to grow, audio compression is an increasingly important problem. In lossy signal compression we aim at minimizing the bitrate of a sample while also minimizing the amount of distortion according to a given metric, ideally correlated with human perception. Audio codecs typically employ a carefully engineered pipeline combining an encoder and a decoder to remove redundancies in the audio content and yield a compact bitstream. Traditionally, this is achieved by decomposing the input with a signal processing transform and trading off the quality of the components that are less likely to influence perception. Leveraging neural networks as trained transforms via an encoder-decoder mechanism has been explored by Morishima et al. (1990); Rippel et al. (2019); Zeghidour et al. (2021). Our research work is in the continuity of this line of work, with a focus on audio signals.

The problems arising in lossy neural compression models are twofold: first, the model has to represent a wide range of signals, such as not to overfit the training set or produce artifact laden audio outside its

---

[*],[†]Equal contribution.

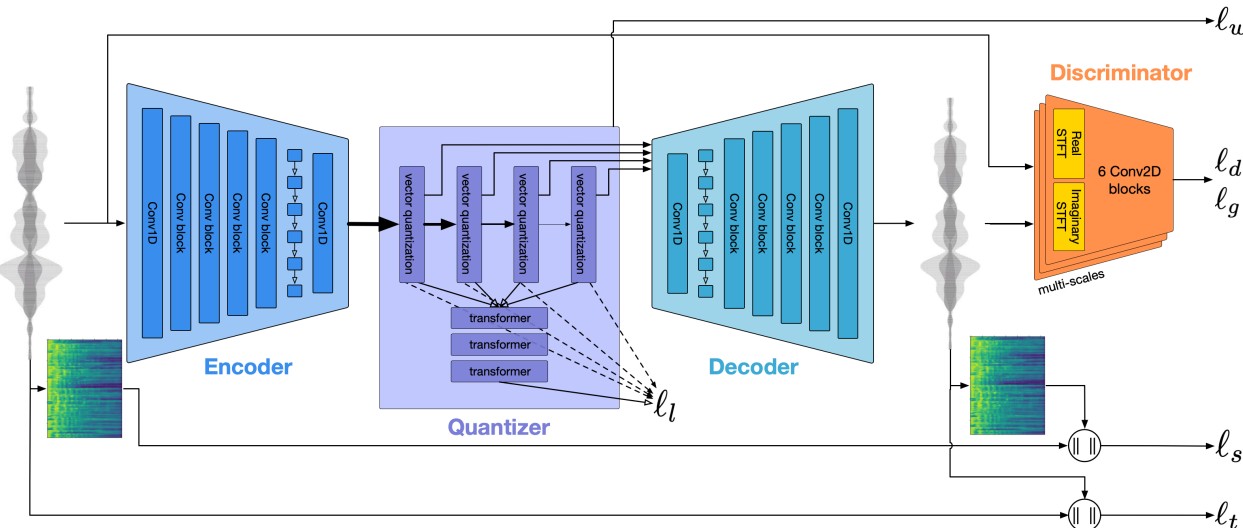

Figure 1: ENCODEC : an encoder decoder codec architecture which is trained with reconstruction ($\ell_s$ and $\ell_t$) as well as adversarial losses ($\ell_g$ for the generator and $\ell_d$ for the discriminator). The residual vector quantization commitment loss ($\ell_w$) applies only to the encoder. Optionally, we train a small Transformer language model for entropy coding over the quantized units with $\ell_l$, which reduces bandwidth even further.

comfort zone. We solve this by having a large and diverse training set (described in Section 4.1), as well as discriminator networks (see Section 3.4) that serve as perceptual losses, which we study extensively in Section 4.5.1, Table 2. The other problem is that of compressing efficiently, both in compute time and in size. For the former, we limit ourselves to models that run in real-time on a single CPU core. For the latter, we use residual vector quantization of the neural encoder floating-point output, for which various approaches have been proposed (Van Den Oord et al., 2017; Zeghidour et al., 2021).

Accompanying those technical contributions, we posit that designing end-to-end neural compression models is a set of intertwined choices, among which at least the encoder-decoder architecture, the quantization method, and the perceptual loss play key parts. Objective evaluations exist and we report scores on them in our ablations (Section 4.5.1). But the evaluation of lossy audio codecs necessarily relies on human perception, so we ran extensive human evaluation for multiple points in this design space, both for speech and music. Those evaluations (MUSHRA) consist in having humans listen to, compare, and rate excerpts of speech or music compressed with competitive codecs and variants of our method, and the uncompressed ground truth. This allows to compare variants of the whole pipeline in isolation, as well as their combined effect, in Section 4.5.1 (Figure 3 and Table 1). Finally, our best model, ENCODEC , reaches state-of-the-art scores for speech and for music at 1.5, 3, 6, 12 kbps at 24 kHz, and at 6, 12, and 24 kbps for 48 kHz with stereo channels.

## 2    Related Work

**Speech and Audio Synthesis.** Recent advancements in neural audio generation enabled computers to efficiently generate natural sounding audio. The first convincing results were achieved by autoregressive models such as WaveNet (Oord et al., 2016), at the cost of slow inference. While many other approaches were explored (Yamamoto et al., 2020a; Kalchbrenner et al., 2018; Goel et al., 2022), the most relevant ones here are those based on Generative Adversarial Networks (GAN) (Kumar et al., 2019; Yamamoto et al., 2020a; Kong et al., 2020; Andreev et al., 2022) were able to match the quality of autoregressive by combining various adversarial networks operate at different multi-scale and multi-period resolutions. Our work uses and extends similar adversarial losses to limit artifacts during audio generation.

**Audio Codec.** Low bitrate parametric speech and audio codecs have long been studied (Atal & Hanauer, 1971; Juang & Gray, 1982), but their quality has been severely limited. Despite some advances (Griffin & Lim, 1985; McCree et al., 1996), modeling the excitation signal has remained a challenging task. The

current state-of-the-art traditional audio codecs are Opus (Valin et al., 2012) and Enhanced Voice Service (EVS) (Dietz et al., 2015). These methods produce high coding efficiency for general audio while supporting various bitrates, sampling rates, and real-time compression.

Neural based audio codecs have been recently proposed and demonstrated promising results (Kleijn et al., 2018; Valin & Skoglund, 2019b; Lim et al., 2020; Kleijn et al., 2021; Zeghidour et al., 2021; Omran et al., 2022; Lin et al., 2022; Jayashankar et al., 2022; Li et al.; Jiang et al., 2022), where most methods are based on quantizing the latent space before feeding it to the decoder. In Valin & Skoglund (2019b), an LPCNet (Valin & Skoglund, 2019a) vocoder was conditioned on hand-crafted features and a uniform quantizer. Gârbacea et al. (2019) conditioned a WaveNet based model on discrete units obtained from a VQ-VAE (Van Den Oord et al., 2017; Razavi et al., 2019) model, while Skoglund & Valin (2019) tried feeding the Opus codec (Valin et al., 2012) to a WaveNet to further improve its perceptual quality. Jayashankar et al. (2022); Jiang et al. (2022) propose an auto-encoder with a vector quantization layer applied over the latent representation and minimizing the reconstruction loss, while Li et al. suggested using Gumbel-Softmax (GS) (Jang et al., 2017) for representation quantization. The most relevant related work to ours is the SoundStream model (Zeghidour et al., 2021), in which the authors propose a fully convolutional encoder decoder architecture with a Residual Vector Quantization (RVQ) (Gray, 1984; Vasuki & Vanathi, 2006) layers. The model was optimized using both reconstruction loss and adversarial perceptual losses. Caillon & Esling (2021) studied compression as part of VAE-based audio modeling , but they did not report any objective or subjective evaluations for this particular application.

**Audio Discretization.** Representing audio and speech using discrete values was proposed to various tasks recently. Dieleman et al. (2018); Dhariwal et al. (2020) proposed a hierarchical VQ-VAE based model for learning discrete representation of raw audio, next combined with an auto-regressive model, demonstrating the ability to generate high quality music. Similarly, Lakhotia et al. (2021); Kharitonov et al. (2021) demonstrated that self-supervised learning methods for speech (e.g., HuBERT (Hsu et al., 2021)), can be quantized and used for conditional and unconditional speech generation. Similar methods were applied to speech resynthesis (Polyak et al., 2021), speech emotion conversion (Kreuk et al., 2021), spoken dialog system (Nguyen et al., 2022), and speech-to-speech translation (Lee et al., 2021a;b; Popuri et al., 2022).

## 3 Model

An audio signal of duration $d$ can be represented by a sequence $\boldsymbol{x} \in [-1, 1]^{C_\mathrm{a} \times T}$ with $C_\mathrm{a}$ the number of audio channels, $T = d \cdot f_\mathrm{sr}$ the number of audio samples at a given sample rate $f_\mathrm{sr}$. The ENCODEC model is composed of three main components: (i) First, an encoder network $E$ is input an audio extract and outputs a latent representation $\boldsymbol{z}$; (ii) Next, a quantization layer $Q$ produces a compressed representation $\boldsymbol{z}_q$, using vector quantization; (iii) Lastly, a decoder network $G$ reconstructs the time-domain signal, $\hat{\boldsymbol{x}}$, from the compressed latent representation $\boldsymbol{z}_q$. The whole system is trained end-to-end to minimize a reconstruction loss applied over both time and frequency domain, together with a perceptual loss in the form of discriminators operating at different resolutions. A visual description of the proposed method can be seen in Figure 1.

### 3.1 Encoder & Decoder Architecture

The ENCODEC model is a simple streaming, convolutional-based encoder-decoder architecture with sequential modeling component applied over the latent representation, both on the encoder and on the decoder side. Such modeling framework was shown to provide great results in various audio-related tasks, e.g., source separation and enhancement (Défossez et al., 2019; Defossez et al., 2020), neural vocoders (Kumar et al., 2019; Kong et al., 2020), audio codec (Zeghidour et al., 2021), and artificial bandwidth extension (Tagliasacchi et al., 2020; Li et al., 2021). We use the same architecture for 24 kHz and 48 kHz audio.

**Encoder-Decoder.** The encoder model $E$ consists in a 1D convolution with $C$ channels and a kernel size of 7 followed by $B$ convolution blocks. Each convolution block is composed of a single residual unit followed by a down-sampling layer consisting in a strided convolution, with a kernel size $K$ of twice the stride $S$. The residual unit contains two convolutions with kernel size 3 and a skip-connection. The number of channels is doubled whenever down-sampling occurred. The convolution blocks are followed by a two-layer LSTM for sequence modeling and a final 1D convolution layer with a kernel size of 7 and $D = 128$ output channels.

---

**Algorithm 1** Residual Vector Quantization (RVQ) algorithm

---

 **procedure** RVQ($\boldsymbol{z}, Q, N_q$):
  $\boldsymbol{z}_q \leftarrow$ Empty List, $\boldsymbol{z}_h \leftarrow 0.0$, $\boldsymbol{r} \leftarrow \boldsymbol{z}$
  **for** $i = 1$ to $N_q$ **do**
   $\hat{\boldsymbol{r}}, r_{\text{idx}} = Q_i(\boldsymbol{r})$           $\triangleright$ Get both codebook dense representation and index
   $\boldsymbol{z}_h = \boldsymbol{z}_h + \hat{\boldsymbol{r}}$
   $\boldsymbol{r} = \boldsymbol{r} - \hat{\boldsymbol{r}}$
   $\boldsymbol{z}_q = \boldsymbol{z}_q \oplus r_{\text{idx}}$               $\triangleright$ Append the codebook index
  **end for**
  **return** $(\boldsymbol{z}_h, \boldsymbol{z}_q)$
 **end procedure**

---

Following Zeghidour et al. (2021); Li et al. (2021), we use $C = 32$, $B = 4$ and (2, 4, 5, 8) as strides. We use ELU as a non-linear activation function (Clevert et al., 2015) either layer normalization (Ba et al., 2016) or weight normalization (Salimans & Kingma, 2016). We use two variants of the model, depending on whether we target the low-latency streamable setup, or a high fidelity non-streamable usage. With this setup, the encoder outputs 75 latent steps per second of audio at 24 kHz, and 150 at 48 kHz. The decoder mirrors the encoder, using transposed convolutions instead of strided convolutions, and with the strides in reverse order as in the encoder, outputting the final mono or stereo audio.

**Non-streamable.** In the non-streamable setup, we use for each convolution a total padding of $K - S$, split equally before the first time step and after the last one (with one more before if $K - S$ is odd). We further split the input into chunks of 1 seconds, with an overlap of 10 ms to avoid clicks, and normalize each chunk before feeding it to the model, applying the inverse operation on the output of the decoder, adding a negligible bandwidth overhead to transmit the scale. We use layer normalization (Ba et al., 2016), computing the statistics including also the time dimension in order to keep the relative scale information.

**Streamable.** For the streamable setup, all padding is put before the first time step. For a transposed convolution with stride $s$, we output the $s$ first time steps, and keep the remaining $s$ steps in memory for completion when the next frame is available, or discarding it at the end of a stream. Thanks to this padding scheme, the model can output 320 samples (13 ms) as soon as the first 320 samples (13 ms) are received. We replace the layer normalization with statistics computed over the time dimension with weight normalization (Salimans & Kingma, 2016), as the former is ill-suited for a streaming setup. We notice a small gain over the objective metrics by keeping a form of normalization, as demonstrated in Table A.3.

### 3.2 Residual Vector Quantization

We use Residual Vector Quantization (RVQ) to quantize the output of the encoder as introduced by Zeghidour et al. (2021). Vector quantization consists in projecting an input vector onto the closest entry in a codebook of a given size. RVQ refines this process by computing the residual after quantization, and further quantizing it using a second codebook, and so forth. A pseudo-code describing the algorithm can be found in Algorithm 1.

We follow the same training procedure as described by Dhariwal et al. (2020) and Zeghidour et al. (2021). The codebook entry selected for each input is updated using an exponential moving average with a decay of 0.99. Following Dhariwal et al. (2020), we track the moving average of the cluster utilization, and clusters whose average size fall below 2 are replaced by a candidate sampled from the current batch. We use a straight-through-estimator (Bengio et al., 2013) to compute the gradient of the encoder, e.g. as if the quantization step was the identity function during the backward phase. Finally, a commitment loss, consisting of the MSE between the input of the quantizer and its output, with gradient only computed with respect to its input, is added to the overall training loss.

By selecting a variable number of residual steps at train time, a single model can be used to support multiple bandwidth target (Zeghidour et al., 2021). For all of our models, we use at most 32 codebooks (16 for the 48 kHz models) with 1024 entries each, e.g. 10 bits per codebook. When doing variable bandwidth training, we select randomly a number of codebooks as a multiple of 2, i.e. corresponding to a bandwidth 1.5, 3, 6, 12

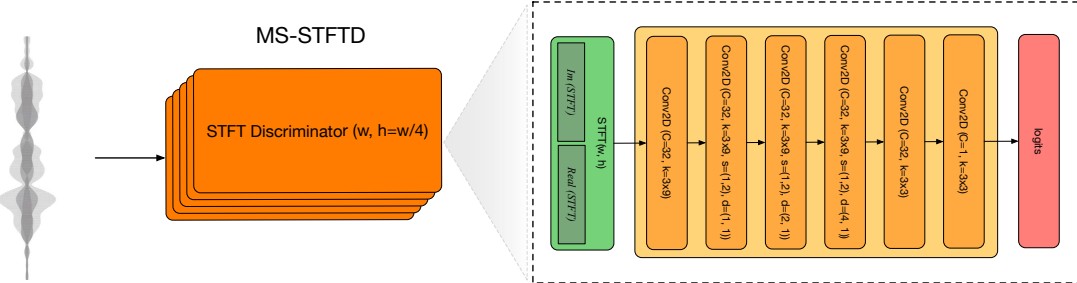

Figure 2: MS-STFT Discriminator architecture. The input to the network is a complex-valued STFT with the real and imaginary parts concatenated. Each discriminator is composed of a 2D convolutional layer, followed by 2D convolutions with increasing dilation rates. Then a final 2D convolution is applied.

or 24 kbps at 24 kHz. Given a continuous latent representation with shape $[B, D, T]$ that comes out of the encoder, this procedure turns it into a discrete set of indexes $[B, N_q, T]$ with $N_q$ the number of codebooks selected. This discrete representation can changed again to a vector by summing the corresponding codebook entries, which is done just before going into the decoder.

### 3.3 Language Modeling and Entropy Coding

Naive coding of the codebook indexes is only optimal if the true underline distribution is uniform over the codebooks. Alternatively, one can better estimate the probability distribution over the codebooks while leveraging past decoded information to improve compression rates. To do so, we train a small Transformer based language model (Vaswani et al., 2017) with the objective of keeping faster than real time end-to-end compression/decompression on a single CPU core. The model consists of 5 layers, 8 heads, 200 channels, a dimension of 800 for the feed-forward blocks, and no dropout. At train time, we select a bandwidth and the corresponding number of codebooks $N_q$. For a time step $t$, the discrete representation obtained at time $t-1$ is transformed into a continuous representation using learnt embedding tables, one for each codebook, and which are summed. For $t = 0$, a special token is used instead. The output of the Transformer is fed into $N_q$ linear layers with as many output channels as the cardinality of each codebook (e.g. 1024), giving us the logits of the estimated distribution over each codebook for time $t$. We thus neglect potential mutual information between the codebooks at a single time step. This allows to speedup inference (as opposed to having one time step per codebook, or a multi-stage prediction) with a limited impact over the final cross entropy. Each attention layer has a causal receptive field of 3.5 seconds, and we offset by a random amount the initial position of the sinusoidal position embedding to emulate being in a longer sequence. We train the model on sequences of 5 seconds.

**Entropy Encoding.** We use a range based arithmetic coder (Pasco, 1976; Rissanen & Langdon, 1981) in order to leverage the estimated probabilities given by the language model. As noted by Ballé et al. (2018), evaluation of the same model might lead to different results on different architectures, or with different evaluation procedures due to floating point approximations. This can lead to decoding errors as the encoder and decoder will not use the exact same code. We observe in particular that the difference between batch evaluation (e.g. all time steps at once), and the real-life streaming evaluation that occurs in the decoder can lead to difference larger than $10^{-8}$. We first round the estimated probabilities with a precision of $10^{-6}$, although evaluations in more contexts would be needed for practical deployment. We use a total range width of $2^{24}$, and assign a minimum range width of 2. We discuss the impact on the processing time in Section 4.6.

### 3.4 Training objective

We detail the training objective that combines a reconstruction loss term, a perceptual loss term (via discriminators), and the RVQ commitment loss.

**Reconstruction Loss.** The reconstruction loss term is comprised of a time and a frequency domain loss term. We minimize the L1 distance between the target and compressed audio over the time domain, i.e.

$\ell_t(\boldsymbol{x}, \hat{\boldsymbol{x}}) = \|\boldsymbol{x} - \hat{\boldsymbol{x}}\|_1$. For the frequency domain, we use a linear combination between the L1 and L2 losses over the mel-spectrogram using several time scales (Yamamoto et al., 2020b; Gritsenko et al., 2020). Formally,

$$\ell_s(\boldsymbol{x}, \hat{\boldsymbol{x}}) = \frac{1}{|\alpha| \cdot |s|} \sum_{\alpha_i \in \alpha} \sum_{i \in e} \|\mathcal{S}_i(\boldsymbol{x}) - \mathcal{S}_i(\hat{\boldsymbol{x}})\|_1 + \alpha_i \|\mathcal{S}_i(\boldsymbol{x}) - \mathcal{S}_i(\hat{\boldsymbol{x}})\|_2, \tag{1}$$

where $\mathcal{S}_i$ is a 64-bins mel-spectrogram using a normalized STFT with window size of $2^i$ and hop length of $2^i/4$, $e = 5, \dots, 11$ is the set of scales, and $\alpha$ represents the set of scalar coefficients balancing between the L1 and L2 terms. Unlike Gritsenko et al. (2020), we take $\alpha_i = 1$.

**Discriminative Loss.** To further improve the quality of the generated samples, we introduce a perceptual loss term based on a multi-scale STFT-based (MS-STFT) discriminator, illustrated in Figure 2. Multi-scale discriminators are popular for capturing different structures in audio signals (Kumar et al., 2019; Kong et al., 2020; You et al., 2021). The basic idea of this family of models is to construct a set of discriminators operating at different scales, which will act as perceptual loss so they can capture different artifacts produced by the generator. Multi-scaling can take place in various shapes. Kumar et al. (2019) proposed a Multi-Scale Discriminator (MSD) which consists of multiple sub-discriminators operating at different scales on the raw waveform (i.e., different window sizes). Kong et al. (2020) proposed a Multi-Period Discriminator (MPD) which consists of multiple sub-discriminators operating on equally spaced samples from the waveform.

Inspired by this line of work, we proposed The MS-STFT discriminator which consists in identically structured networks operating on multi-scaled complex-valued STFT with the real and imaginary parts concatenated. Each sub-network is composed of a 2D convolutional layer (using kernel size 3 x 8 with 32 channels), followed by 2D convolutions with increasing dilation rates in the time dimension of 1, 2 and 4, and a stride of 2 over the frequency axis. A final 2D convolution with kernel size 3 x 3 and stride (1, 1) provide the final prediction. We use 5 different scales with STFT window lengths of [2048, 1024, 512, 256, 128]. For 48 kHz audio, we double the size of each STFT window and train the discriminator every two batches, and for stereophonic audio, we process separately the left and right channels. We use LeakyReLU as a non-linear activation function and apply weight normalization (Salimans & Kingma, 2016) to our discriminator network. The MS-STFT discriminator model architecture is visually depicted in Figure 2.

The adversarial loss for the generator is constructed as follows, $\ell_g(\hat{\boldsymbol{x}}) = \frac{1}{K} \sum_k \max(0, 1 - D_k(\hat{\boldsymbol{x}}))$, where $K$ is the number of discriminators. Similarly to previous work on neural vocoders (Kumar et al., 2019; Kong et al., 2020; You et al., 2021), we additionally include a relative feature matching loss for the generator. Formally,

$$\ell_{feat}(\boldsymbol{x}, \hat{\boldsymbol{x}}) = \frac{1}{KL} \sum_{k=1}^{K} \sum_{l=1}^{L} \frac{\|D_k^l(\boldsymbol{x}) - D_k^l(\hat{\boldsymbol{x}})\|_1}{\mathrm{mean}\left(\|D_k^l(\boldsymbol{x})\|_1\right)}, \tag{2}$$

where the mean is computed over all dimensions, $(D_k)$ are the discriminators, and $L$ is the number of layers in discriminators. The discriminators are trained to minimize the following hinge-loss adversarial loss function: $L_d(\boldsymbol{x}, \hat{\boldsymbol{x}}) = \frac{1}{K} \sum_{k=1}^{K} \max(0, 1 - D_k(\boldsymbol{x})) + \max(0, 1 + D_k(\hat{\boldsymbol{x}}))$, where $K$ is the number of discriminators. Given that the discriminator tend to overpower easily the decoder, we update its weight with a probability of 2/3 at 24 kHz, and 0.5 at 48 kHz.

**Multi-bandwidth training.** At 24 kHz, we train the model to support the bandwidths 1.5, 3, 6, 12, and 24 kbps by selecting the appropriate number of codebooks to keep in the RVQ step, as explained in Section 3.2. At 48 kHz, we train to support 3, 6, 12 and 24 kbps. We also noticed that using a dedicated discriminator per-bandwidth is beneficial to the audio quality. Thus, we select a given bandwidth for the entire batch, and evaluate and update only the corresponding discriminator.

**VQ commitment loss.** As mentioned in Section 3.2, we add a commitment loss $l_w$ between the output of the encoder, and its quantized value, with no gradient being computed for the quantized value. For each residual step $c \in \{1, \dots C\}$ (with $C$ depeding on the bandwidth target for the current batch), noting $\boldsymbol{z}_c$ the current residual and $q_c(\boldsymbol{z}_c)$ the nearest entry in the corresponding codebook, we define $l_w$ as

$$l_w = \sum_{c=1}^{C} \|\boldsymbol{z}_c - q_c(\boldsymbol{z}_c)\|_2^2. \tag{3}$$

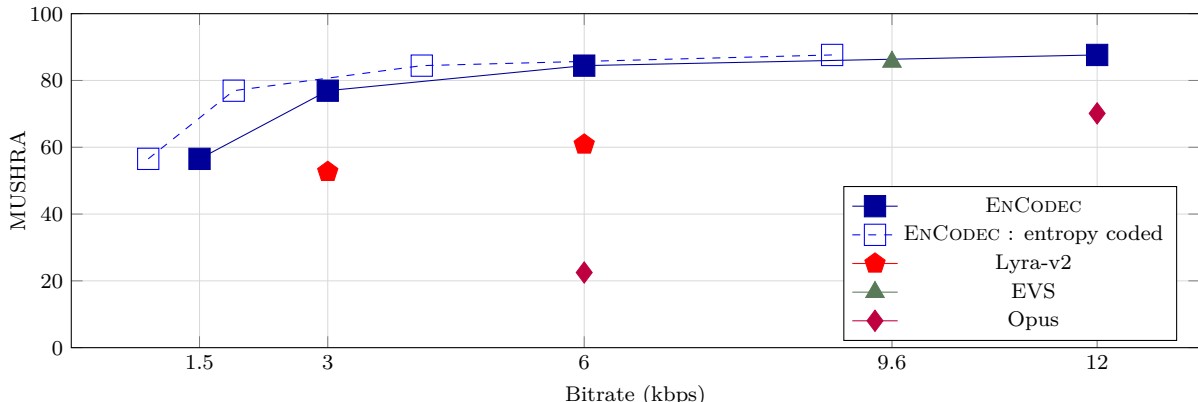

Figure 3: Human evaluations (MUSHRA: comparative scoring of samples) across bandwidths of standard codecs and neural codecs. For ENCODEC we report the initial bandwidth without entropy coding (in plain) and with entropy coding (hollow). Lyra-v2 is a neural audio codec, while EVS and Opus are competitive standard codecs. The audio samples are from speech and music. The ground truth is 16bits 24kHz wave.

Overall, the generator is trained to optimize the following loss, summed over the batch,

$$L_G = \lambda_t \cdot \ell_t(\boldsymbol{x}, \hat{\boldsymbol{x}}) + \lambda_s \cdot \ell_s(\boldsymbol{x}, \hat{\boldsymbol{x}}) + \lambda_g \cdot \ell_g(\hat{\boldsymbol{x}}) + \lambda_{feat} \cdot \ell_{feat}(\boldsymbol{x}, \hat{\boldsymbol{x}}) + \lambda_w \cdot \ell_w(w), \tag{4}$$

where $\lambda_t$, $\lambda_s$, $\lambda_g$, $\lambda_{feat}$, and $\lambda_w$ the scalar coefficients to balance between the terms.

**Balancer.** We introduce a loss balancer in order to stabilize training, in particular the varying scale of the gradients coming from the discriminators. We also find that the balancer makes it easier to reason about the different loss weights, independently of their scale. Let us take a number of losses $(\ell_i)_i$ that depends only on the output of the model $\hat{x}$. We define $g_i = \frac{\partial \ell_i}{\partial \hat{x}}$, and $\langle \|g_i\|_2 \rangle_\beta$ the exponential moving average of $g_i$ over the last training batches. Given a set of weights $(\lambda_i)$ and a reference norm $R$, we define

$$\tilde{g}_i = R \frac{\lambda_i}{\sum_j \lambda_j} \cdot \frac{g_i}{\langle \|g_i\|_2 \rangle_\beta}. \tag{5}$$

We then backpropagate into the network $\sum_i \tilde{g}_i$, instead of the original $\sum_i \lambda_i g_i$. This changes the optimization problem but allows to make the $\lambda_i$ interpretable irrespectively of the natural scale of each loss. If $\sum_i \lambda_i = 1$, then each weight can be interpreted as the fraction of the model gradient that come from the corresponding loss. We take $R = 1$ and $\beta = 0.999$. All the generator losses from Eq. (4) fit into the balancer, except for the commitment loss, as it is not defined with respect to the output of the model.

## 4 Experiments and Results

### 4.1 Dataset

We train ENCODEC on 24 kHz monophonic across diverse domains, namely: speech, noisy speech, music and general audio while we train the fullband stereo ENCODEC on only 48 kHz music. For speech, we use the clean speech segments from DNS Challenge 4 (Dubey et al., 2022) and the Common Voice dataset (Ardila et al., 2019). For general audio, we use on AudioSet (Gemmeke et al., 2017) together with FSD50K (Fonseca et al., 2021). For music, we rely on the Jamendo dataset (Bogdanov et al., 2019) for training and evaluation and we further evaluate our models on music using a proprietary music dataset. Data splits are detailed in Appendix A.1.

For training and validation, we define a mixing strategy which consists in either sampling a single source from a dataset or performing on the fly mixing of two or three sources. Specifically, we have four strategies: (s1) we sample a single source from Jamendo with probability 0.32; (s2) we sample a single source from the other datasets with the same probability; (s3) we mix two sources from all datasets with a probability of 0.24; (s4) we mix three sources from all datasets except music with a probability of 0.12.

The audio is normalized by file and we apply a random gain between -10 and 6 dB. We reject any sample that has been clipped. Finally we add reverberation using room impulse responses provided by the DNS challenge with probability 0.2, and RT60 in the range [0.3, 1.3] except for the single-source music samples. Such input normalization was found to be beneficial in improving model performance (Defossez et al., 2020). For testing, we use four categories: clean speech from DNS alone, clean speech mixed with FSDK50K sample, Jamendo sample alone, proprietary music sample alone.

### 4.2 Baselines

Opus (Valin et al., 2012) is a versatile speech and audio codec standardized by the IETF in 2012. It scales from 6 kbps narrowband monophonic audio to 510 kbps fullband stereophonic audio. EVS (Dietz et al., 2015) is a codec standardized in 2014 by 3GPP and developed for Voice over LTE (VoLTE). It supports a range of bitrates from 5.9 kbps to 128 kbps, and audio bandwidths from 4 kHz to 20 kHz. It is the successor of AMR-WB (Bhagat et al., 2012).We use both codecs to serve as traditional digital signal processing baselines. We also utilize MP3 compression at 64 kbps as an additional baseline for the stereophonic signal compression case. MP3 uses lossy data compression by approximating the accuracy of certain components of sound that are considered to be beyond hearing capabilities of most humans. Finally, we compare ENCODEC to the SoundStream model from the official implementation available in Lyra 2 [1] at 3.2 kbps and 6 kbps on audio upsampled to 32 kHz. We also reproduced a version of SoundStream (Zeghidour et al., 2021) with minor improvements. Namely, we use the relative feature loss introduce in Section 3.4, and layer normalization (applied separately for each time step) in the discriminators, except for the first and last layer, which improved the audio quality during our preliminary studies. Results a reported in Table A.2 in the Appendix A.3.

### 4.3 Evaluation Methods

We consider subjective and objective metrics. For the subjective tests we follow the MUSHRA protocol (Series, 2014), using a hidden reference and a low anchor. Annotators were recruited using a crowd-sourcing platform, in which they were asked to rate the perceptual quality of the provided samples in a range between 1 to 100. We randomly select 50 samples of 5 seconds from each category of the the test set and force at least 10 annotations per samples. To filter noisy annotations and outliers we remove annotators who rate the reference recordings less then 90 in at least 20% of the cases, or rate the low-anchor recording above 80 more than 50% of the time. For objective metrics, we use ViSQOL (Hines et al., 2012; Chinen et al., 2020) [2], together with the Scale-Invariant Signal-to-Noise Ration (SI-SNR) (Luo & Mesgarani, 2019; Nachmani et al., 2020; Chazan et al., 2021).

### 4.4 Training

We train all models for 300 epochs, with one epoch being 2,000 updates with the Adam optimizer with a batch size of 64 examples of 1 second each, a learning rate of $3 \cdot 10^{-4}$, $\beta_1 = 0.5$, and $\beta_2 = 0.9$. All the models are traind using 8 A100 GPUs. We use the balancer introduced in Section 3.4 with weights $\lambda_t = 0.1$, $\lambda_f = 1$, $\lambda_g = 3$, $\lambda_{\text{feat}} = 3$ for the 24 kHz models. For the 48 kHz model, we use instead $\lambda_g = 4$, $\lambda_{\text{feat}} = 4$.

### 4.5 Results

We start with the results for ENCODEC with a bandwidth in $\{1.5, 3, 6, 12\}$ kbps and compare them to the baselines. Results for the streamable setup are reported in Figure 3 and a breakdown per category in Table 1. We additionally explored other quantizers such as Gumbel-Softmax and DiffQ (see details in Appendix A.2), however, we found in preliminary results that they provide similar or worse results, hence we do not report them.

When considering the same bandwidth, ENCODEC is superior to all evaluated baselines considering the MUSHRA score. Notice, ENCODEC at 3kbps reaches better performance on average than Lyra-v2 using 6kbps and Opus at 12kbps. When considering the additional language model over the codes, we can reduce the bandwidth by $\sim 25 - 40\%$. For instance, we can reduce the bandwidth of the 3 kpbs model to 1.9 kbps.

---

[1] https://github.com/google/lyra
[2] We compute visqol with: `https://github.com/google/visqol` using the recommended recipes.

Table 1: MUSHRA scores for Opus, EVS, Lyra-v2, and ENCODEC for various bandwidths under the streamable setting. Results are reported across different audio categories (clean speech, noisy speech, and music), sampled at 24 kHz. We report mean scores and 95% confidence intervals. For ENCODEC, we also report the average bandwidth after using the entropy coding described in Section 3.3.

| Model | Bandwidth | Entropy Coded | Clean Speech | Noisy Speech | Music Set-1 | Music Set-2 |
|---|---|---|---|---|---|---|
| Reference | - | - | $95.5_{\pm1.6}$ | $93.9_{\pm1.8}$ | $93.2_{\pm2.5}$ | $97.1_{\pm1.3}$ |
| Opus | 6.0 kbps | - | $30.1_{\pm2.8}$ | $19.1_{\pm5.9}$ | $20.6_{\pm5.8}$ | $17.9_{\pm5.3}$ |
| Opus | 12.0 kbps | - | $76.5_{\pm2.3}$ | $61.9_{\pm2.1}$ | $77.8_{\pm3.2}$ | $65.4_{\pm2.7}$ |
| EVS | 9.6 kbps | - | $84.4_{\pm2.5}$ | $80.0_{\pm2.4}$ | $89.9_{\pm2.3}$ | $87.7_{\pm2.3}$ |
| Lyra-v2 | 3.0 kbps | - | $53.1_{\pm1.9}$ | $52.0_{\pm4.7}$ | $69.3_{\pm3.3}$ | $42.3_{\pm3.5}$ |
| Lyra-v2 | 6.0 kbps | - | $66.2_{\pm2.9}$ | $59.9_{\pm3.3}$ | $75.7_{\pm2.6}$ | $48.6_{\pm2.1}$ |
| ENCODEC | 1.5 kbps | 0.9 kbps | $49.2_{\pm2.4}$ | $41.3_{\pm3.6}$ | $68.2_{\pm2.2}$ | $66.5_{\pm2.3}$ |
| ENCODEC | 3.0 kbps | 1.9 kbps | $67.0_{\pm1.5}$ | $62.5_{\pm2.3}$ | $89.6_{\pm3.1}$ | $87.8_{\pm2.9}$ |
| ENCODEC | 6.0 kbps | 4.1 kbps | $83.1_{\pm2.7}$ | $69.4_{\pm2.3}$ | $92.9_{\pm1.8}$ | $91.3_{\pm2.1}$ |
| ENCODEC | 12.0 kbps | 8.9 kbps | $90.6_{\pm2.6}$ | $80.1_{\pm2.5}$ | $91.8_{\pm2.5}$ | $92.9_{\pm1.2}$ |

Table 2: Comparing discriminators using objective (ViSQOL, SI-SNR) and subjective metrics (MUSHRA).

| Discriminator setup | SI-SNR | ViSQOL | MUSHRA |
|---|---|---|---|
| MSD+Mono-STFT | 5.99 | 4.22 | $62.91_{\pm2.62}$ |
| MPD | 7.35 | 4.24 | $60.7_{\pm2.8}$ |
| MS-STFT+MPD | 6.55 | **4.34** | $\mathbf{79.0_{\pm1.9}}$ |
| MS-STFT | **6.67** | **4.35** | $\mathbf{77.5_{\pm1.8}}$ |

We observe that for higher bandwidth, the compression ratio is lower, which could be explained by the small size of the Transformer model used, making hard to model all codebooks together.

### 4.5.1 Ablation study

Next, we perform an ablation study to better evaluate the effect of the discriminator setup, streaming, multi-target bandwidth, and balancer. We provide more detailed ablation studies in the Appendix, Section A.3.

**The effect of discriminators setup.** Various discriminators were proposed in prior work to improve the perceptual quality of the generated audio. The Multi-Scale Discriminator (MSD) model proposed by Kumar et al. (2019) and adopted in (Kong et al., 2020; Andreev et al., 2022; Zeghidour et al., 2021), operates on the raw waveform at different resolutions. We adopt the same MSD configuration as described in Zeghidour et al. (2021). Kong et al. (2020) additionally propose the Multi-Period Discriminator (MPD) model, which reshapes the waveform to a 2D input with multiple periods. Next, the STFT Discriminator (Mono-STFTD) model was introduced in Zeghidour et al. (2021), where a single network operates over the complex-valued STFT.

We evaluate our MS-STFTD discriminator against three other discriminator configurations: (i) MSD+Mono-STFTD (as in Zeghidour et al. (2021)); (ii) MPD only; (iii) MS-STFTD only; (vi) MS-STFTD+MPD. Results are reported in Table 2. Results suggest that using only a multi-scale STFT-based discriminator such as MS-STFTD, is enough to generate high quality audio. Additionally, it simplifies the model training and reduces training time. Including the MPD discriminator, adds a small gain when considering the MUSHRA score.

**The effect of the streamable modeling.** We also investigate streamable vs. non-streamable setups and report results in Table 3. Unsurprisingly, we notice a small degradation switching from non-streamable to streamable but the performance remains strong while this setting enables streaming inference.

Table 3: Streamable vs. Non-streamable evaluations at 6 kbps on an equal mix of speech and music.

| Model | Streamable | SI-SNR | ViSQOL |
|-------|:----------:|:------:|:------:|
| Opus | ✓ | 2.45 | 2.60 |
| EVS | ✓ | 1.89 | 2.74 |
| ENCODEC | ✓ | 6.67 | 4.35 |
| ENCODEC | ✗ | 7.46 | 4.39 |

Table 4: **Stereophonic** extreme music compression versus MP3 and Opus for music sampled at 48 kHz.

| Model | Bandwidth | Entropy Coded | Compression | MUSHRA |
|-------|:---------:|:-------------:|:-----------:|:------:|
| Reference | - | - | $1\times$ | $\mathbf{95.1_{\pm1.8}}$ |
| MP3 | 64 kbps | - | $24\times$ | $82.7_{\pm3.2}$ |
| Opus | 6 kbps | - | $256\times$ | $17.7_{\pm5.9}$ |
| Opus | 24 kbps | - | $64\times$ | $82.9_{\pm3.7}$ |
| ENCODEC | 6 kbps | 4.2 kbps | $256\times$ | $82.9_{\pm2.4}$ |
| ENCODEC | 12 kbps | 8.9 kbps | $128\times$ | $\mathbf{88.0_{\pm2.7}}$ |
| ENCODEC | 24 kbps | 19.4 kbps | $64\times$ | $\mathbf{87.5_{\pm2.6}}$ |

**The effect of the balancer.** Lastly, we present results evaluating the impact of the balancer. We train the ENCODEC model considering various values $\lambda_t$, $\lambda_f$, $\lambda_g$, and $\lambda_{feat}$ with and without the balancer. Results are reported in Table A.4 in the Appendix. As expected, results suggest the balancer significantly stabilizes the training process. See Appendix A.3 for more details.

### 4.5.2 Stereo Evaluation

All previously reported results considered only the monophonic setup. Although it makes sense when considering speech data, however for music data, stereo compression is highly important. We adjust our current setup to stereo by only modifying our discriminator setup as described in Section 3.4.

Results for ENCODEC working at 6 kbps, ENCODEC with Residual Vector Quantization (RVQ) at 6 kbps, and Opus at 6 kbps, and MP3 at 64 kbps are reported in Table 4. ENCODEC is significantly outperforms Opus at 6kbps and is comparable to MP3 at 64kbps, while ENCODEC at 12kpbs achieve comparable performance to ENCODEC at 24kbps. Using a language model and entropy coding gives a variable gain between 20% to 30%.

### 4.6 Latency and computation time

We report the initial latency and real time factor on Table 5. The real-time factor is here defined as the ratio between the duration of the audio and the processing time, so that it is greater than one when the method is faster than real time. We profiled all models on a single thread of a MacBook Pro 2019 CPU at 6 kbps.

Table 5: Initial latency and real time factor (RTF) for Lyra v2, ENCODEC at 24 kHz and 48 kHz. A RTF greater than 1 indicates faster than real time processing. We report the RTF for both the encoding (Enc.) and decoding (Dec.), without and with entropy coding (EC). All models are evaluated at 6 kbps.

| Model | Latency | Real Time Factor | | | |
|-------|:-------:|:----:|:----:|:----------:|:----------:|
| | | *Enc.* | *Dec.* | *Enc. + EC* | *Dec. + EC* |
| Lyra v2 (32 kHz) | - | 27.4 | 67.2 | - | - |
| ENCODEC 24 kHz | 13 ms | 9.8 | 10.4 | 1.6 | 1.6 |
| ENCODEC 48 kHz | 1 s | 6.8 | 5.1 | 0.68 | 0.66 |

**Initial latency.** The 24 kHz streaming ENCODEC model has an initial latency (i.e., without the computation time) of 13.3 ms. The 48 kHz non-streaming version has an initial latency of 1 second, due to the normalizations used. Note that using entropy coding increases the initial latency, because the stream cannot be "flushed" with each frame, in order to keep the overhead small. Thus decoding the frame at time $t$, requires for the frame $t + 1$ to be partially received, increasing the latency by 13ms.

**Real time factor.** While our model is worse than Lyra v2 in term of processing speed, it processes the audio 10 times faster than real time, making it a good candidate for real life applications. The gain from the entropy coding comes at a cost, although the processing is still faster than real time and could be used for applications where latency is not essential (e.g. streaming). At 48 kHz, the increased number of step size lead to a slower than real time processing, although a more efficient implementation, or using accelerated hardware would improve the RTF. It could also be used for archiving where real time processing is not required.

## 5  Conclusion

We presented ENCODEC : a state-of-the-art real-time neural audio compression model, producing high-fidelity audio samples across a range of sample rates and bandwidth. We showed subjective and objective results from 24kHz monophonic at 1.5 kbps (Figure 3) to 48kHz stereophonic (Table 4). We improved sample quality by developing a simple but potent spectrogram-only adversarial loss which efficiently reduces artifacts and produce high-quality samples. Besides, we stabilized training and improved the interpretability of the weights for losses through a novel gradient balancer. Finally, we also demonstrated that a small Transformer model can be used to further reduce the bandwidth by up to 40% without further degradation of quality, in particular for applications where low latency is not essential (e.g. music streaming).

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

Table A.1: Datasets description. License with asterisk annotation * imply that the specific license varies across the dataset and is specific to each sample.

| Dataset | Audio domain | Sampling rate | Channels | Duration | License |
|---|---|---|---|---|---|
| Common Voice 7.0 | Speech | 48 kHz | 1 | 9,096 h | CC-0 |
| DNS Challenge 4 (speech) | Speech | 48 kHz | 1 | 2,425 h | Multiples* |
| AudioSet | General audio | 48 kHz | 2 | 4,989 h | CC BY 4.0* |
| FSD50K | General audio | 44.1 kHz | 1 | 108 h | CC* |
| Jamendo | Music | multiples | 2 | 919 h | CC* |

# A  Appendix

## A.1  Experimental details

**Datasets details.** We present additional details and statistics over the datasets used for training in Table A.1. Notice that for some datasets, each sub-source or sample has its own license and we then refer the reader to the respective dataset details for more information. We create our datasets splits as followed. For Common Voice, we randomly sample 99.5% of the dataset for train, 0.25% for valid and the rest for test splits. Similarly, we sample 98% of the clean segments from DNS Challenge 4 for train, 1% for valid and 1% for test. For AudioSet, we use the unbalanced train segments as training data and randomly selected half of the eval segments as validation set and the other half as test set. We follow the same procedure for FSD50K using the dev set for training and splitting the eval set between validation and test. Finally for the Jamendo dataset, we randomly take 96% of the artists and their corresponding tracks for train, 2% for valid and 2% for test, hence there is no artists overlap in the different sets.

**SoundStream model.** We additionally re-implemented SoundStream. We follow the implementation details in Zeghidour et al. (2021) to develop our own SoundStream version as the original implementation is not open sourced. We implement Residual Vector Quantization with k-means initialization, exponential moving average updates and random restart as pointed by the original implementation. For the wave-based discriminator, we follow the details provided in Zeghidour et al. (2021) and refer to the original Multi-Scale Discriminator implementation proposed in Kumar et al. (2019) for additional information. We use LeakyReLU as non-linear activation function and following Kong et al. (2020), we use spectral normalization for the original resolution and weight normalization for other resolutions. For the STFT-based discriminator, we experimented with multiple normalization methods and found that using LayerNorm (Ba et al., 2016) was the only one that prevented the discriminator from diverging. We used LeakyReLU as non-linear activation function. Finally, training hyper parameters are not shared either so we use the same parameters as for our ENCODEC model.

## A.2  Alternative quantizers

### A.2.1  DiffQ Quantizer

**Pseudo quantization noise.** We perform scalar quantization of the latent representation. As quantization is non differentiable, we use properly scaled additive independent noise, a.k.a pseudo quantization noise, at train time to simulate it. This approach was first use for analog-to-digital converters design (Widrow et al., 1996), then for image compression (Ballé et al., 2017), and finally by the DiffQ model compression method (Défossez et al., 2021) with a differentiable bandwidth estimate. We extend the DiffQ approach for latent space quantization, adding support for streamable rescaling, proper sparsity, and improved prior coding. Formally, we introduce a learnt parameter $B \in \mathbb{R}^D$ (with $D$ the dimension of the latent space) such that $B^{(i)}$ represents the number of bits to use of the $i$-th dimension. In practice $B$ is parameterized as $B = B_{\max} \cdot \text{sigmoid}(\alpha v)$, with $B_{\max} = 15$, and the $\alpha = 5$ factor is used for boosting the learning rate of $v$, the learnt parameter. We then define the pseudo quantized representation $z_{q,\text{train}}$ used at train time,

$$z_{q,\text{train}} = \text{clamp}(z, m - L \cdot \sigma, m + L \cdot \sigma) + L \cdot \sigma \cdot \frac{\mathcal{U}[-1,1]}{2^B}, \tag{6}$$

with $m$ (resp. $\sigma$) the mean (resp. standard deviation) of $z$ along the time and batch axis, and $\mathcal{U}[-1,1]$ uniform i.i.d noise. The limit $L$ is chosen so that when $B$ goes to 0, the noise covers most of the range of values accessible to a Gaussian variable of variance $\sigma$. In order to prevent outlier values, we clamp the input $z$ to this expected range of values. If $L$ is too large, the dynamic range of the quantization will be poorly used. If $L$ is too low, many values will get clamped and lose gradients. In practice we choose $L = 3$, which verifies that values are not clamped 99.5% of the time (against 99.8% if the input were gaussian). In order to learn $B$, we approximate at train time the bandwidth used by the model by $w_{\text{diffq}} = T' \sum_{i=1}^{D} B^{(i)}/d$ with $T'$ the number of latent time steps, $d$ the duration of the sample, and add to the training loss a penalty term of the form $\lambda w_{\text{diffq}}$, as long as $w_{\text{diffq}}$ is over a given target, as used in Section 3.4.

**Test time quantization.** At validation and test time, we replace the batch-level statistic with an exponential moving average with a decay of 0.9 computed over the train set, similar to batch norm (Ioffe & Szegedy, 2015). We first normalize and clamp $z$ to the segment $[0,1]$ as $u = \text{clamp}((L + \sigma^{-1}(z - m))/2L, 0, 1)$ and define the number of quantized levels $N_B = \text{round}(2^B)$ and the quantized index as

$$i = \min\left[\text{floor}(N_B \cdot u), N_B - 1\right] \in [0..N_B - 1], \tag{7}$$

with the minimum taken to avoid the edge case $u = 1$. We know we can code each entry in $i$ on at most $\log_2(N_B)$ bits. The quantized latent $z_q$ is finally defined as $z_q = m + L\sigma\left(2\frac{i+0.5}{N_B} - 1\right)$.

**Sparsity.** We want to allow $B$ to go to 0, however additive noise fails to remove all information contained in the latent in that case. Indeed, if $z < m$ for instance, then even after adding the largest possible amount of noise, $z_{q,\text{train}}$ is still biased towards values smaller than $m$. In order to remove all information about $z$ from $z_{q,\text{train}}$, the scale of the noise relative to the scale of the signal must go to infinity. This is achieved by scaling down $z$ by a factor $\min(B, 1)$ in Eq. (6), while scaling down the additive noise only by a factor $\sqrt{\min(B, 1)}$. Thus, the decoder cannot invert the downscaling of $z$ without blowing up the noise. In the limit of $B \to 0$, we recover a sparse representation.

### A.2.2 Gumbel softmax quantizer

We introduce a second fully differentiable vector quantizer composed of $N_C$ codebooks each with $\Omega$ entries. The $i$-th codebook is composed of a set of centroids $C_i \in \mathbb{R}^{\Omega \times D}$, a logit bias $b_i \in \mathbb{R}^\Omega$, and a learnt prior logit $l_i \in \mathbb{R}^\Omega$. Assuming for simplicity a latent vector for the $j$-th time step $z_j \in \mathbb{R}^D$, we define a probability distribution over each codebook entries as $q_i(z) = \text{softmax}(C_i z_j + b_i)$. We then sample from the corresponding gumbel-softmax (Jang et al., 2017) with a temperature $\tau = 0.5$. This gives us a differentiable approximately 1-hot vector over the codebooks, i.e., noting GS the gumbel-softmax,

$$z_{q,\text{train}} = \sum_{i=1}^{N_C} \text{GS}(\log(q_i(z)), \tau)^T C_i. \tag{8}$$

At test time, we replace the gumbel-softmax with a sampling from the distribution $q_i$. We define for all $i$, $p_i = \text{softmax}(l_i)$ the prior distribution over the codebooks entries which is used for coding the quantized value $z_q$ with an arithmetic coder. We can both train the prior and minimize the bandwidth with a single loss term given by the cross entropy between the prior $p_i$ and the posterior $q_i(z)$, i.e. the differentiable bandwidth estimate for this method is given by $w_{\text{gs}} = \sum_{i=1}^{N_C} \sum_{k=1}^{\Omega} -q_i(z) \log(p_i)$.

### A.3 Additional Results

**Comparing to SoundStream.** For fair evaluation, we also compare ENCODEC to our reimplementation of SoundStream (Zeghidour et al., 2021). For which, the quantizer corresponds to RVQ in and the discriminator corresponds to STFTD+MSD. Results are reported in Table A.2. The results of our SoundStream implementation are slightly worse than ENCODEC using DiffQ quantizer. However, when considering the RVQ as the latent quantizer, ENCODEC is superior to the SoundStream model. Both ENCODEC and SoundStream methods are significantly better than Opus and EVS at 6.0kbps.

**The effect of the model architecture.** We investigate the impact of different architectural decisions of our ENCODEC model and we present our results with objective metrics and real-time factor in Table A.3.

Table A.2: A comparison between ENCODEC using either DiffQ or RVQ as latent quantizers against our implementation of the SoundStream model at 3kbps. We additionally include the results of Opus and EVS at 6.0kbps for reference.

| Model | Bandwidth | MUSHRA |
|---|---|---|
| Reference | - | $96.1_{\pm1.41}$ |
| Opus | 6.0 | $21.1_{\pm2.62}$ |
| EVS | 6.0 | $62.9_{\pm2.18}$ |
| SoundStream | 3.0 | $71.8_{\pm1.51}$ |
| ENCODEC (DiffQ) | 3.0 | $72.3_{\pm1.18}$ |
| ENCODEC (RVQ) | 3.0 | $\mathbf{76.8}_{\pm\mathbf{1.31}}$ |

Table A.3: Model architecture Analysis. We explore variations of our architecture including the impact of the number of Residual Units (ResUnits), the sequence modeling with LSTM, the number of channels (C) on objective metrics and real-time factor (RTF greater than 1 is faster than real time).

| Model | RTF Enc | RTF Dec | SI-SNR | ViSQOL |
|---|---|---|---|---|
| ENCODEC base | 9.8 | 10.4 | 6.67 | 4.35 |
| Channels = 16 | 26.0 | 25.7 | 6.40 | 4.32 |
| Channels = 64 | 1.3 | 3.1 | 6.70 | 4.38 |
| norm = None | 10.1 | 10.4 | 6.45 | 4.29 |
| LSTM = 0 | 15.0 | 14.6 | 6.40 | 4.35 |
| Residual Layer = 3, LSTM = 0 | 6.0 | 7.3 | 6.32 | 4.35 |

For all models, we consider the streamable setup and our reference ENCODEC base model has the number of channels $C$ set to 32 and a single residual unit. The real-time factor reported here is defined as the ratio between the duration of the input audio and the processing time needed for encoding (respectively decoding) it. We profiled streamable multi-target 24 kHz models during inference at 6 kbps on a single thread of a MacBook Pro 2019 CPU. First, we validate that the selected number of channels provide good trade offs in terms of perceived quality and inference speed. We observe that increasing the capacity of the model only marginally affects the scores on objective metrics while it has a high impact on the real-time factor. The results also demonstrate that presence of LSTM improves the SI-SNR and the final reconstruction quality, at the detriment of the real-time factor. We also experiment with increasing the number of residual units instead of relying on a LSTM in our architecture. To do so, we use 3 Residual Units and we double the dilation used in the first convolutional layer of the residual unit for each subsequent unit. We observe that using Residual Units has more impact on the real-time factor and we note a small degradation of the SI-SNR compared to the LSTM-based version.

**The effect of the balancer.** Lastly, we present results evaluating the impact of the balancer. In which, we train the ENCODEC model considering various levels of balancing the loss. For this set of experiments we use the DiffQ quantizer rather than RVQ. All models were trained on Jamando music dataset (see Table A.4). Results suggest the balancer significantly stabilizes the training process. This is especially useful while considering the different terms in the objective together where each term operates at a different scale. Following the balancer approach significantly reduce the effort needed for tuning the objective coefficients (i.e., $\lambda_t$, $\lambda_f$, $\lambda_g$, $\lambda_{feat}$). We demonstrate that the use of the balancer shows no degradation compared to an identified combination of coefficients.

## A.4 Societal impact

The majority of the internet traffic is represented by audio and video streams (82% in 2021 according to Cisco (2021)). This share of content is boosted by user-generated content, phone and video calls and by the

Table A.4: ViSQOL and SI-SNR results for ENCODEC using DiffQ considering various coefficients to balance the overall objective. All models were trained using Jamando music dataset.

| $\lambda_t$ | $\lambda_f$ | $\lambda_g$ | $\lambda_{feat}$ | Balancer | SI-SNR | ViSQOL |
|---|---|---|---|---|---|---|
| 1 | 1 | 1 | 1 | ✓ | **10.32** | **4.16** |
| 1 | 1 | 1 | 1 | ✗ | 6.16 | 3.89 |
| 1 | 1 | 2 | 1 | ✓ | **10.08** | **4.12** |
| 1 | 1 | 2 | 1 | ✗ | 5.01 | 3.77 |
| 1 | 2 | 2 | 1 | ✓ | **10.06** | **4.17** |
| 1 | 2 | 2 | 1 | ✗ | 3.84 | 3.67 |
| 1 | 2 | 4 | 1 | ✓ | **9.93** | **4.17** |
| 1 | 2 | 4 | 1 | ✗ | 1.72 | 3.52 |
| 1 | 2 | 100 | 1 | ✓ | **8.41** | **4.05** |
| 1 | 2 | 100 | 1 | ✗ | -35.83 | 2.82 |
| 2 | 1 | 1 | 4 | ✓ | **10.53** | **4.06** |
| 2 | 1 | 1 | 4 | ✗ | 7.66 | 4.03 |
| 2 | 2 | 2 | 4 | ✓ | **10.19** | **4.13** |
| 2 | 2 | 2 | 4 | ✗ | 7.16 | 3.98 |
| 2 | 2 | 10 | 4 | ✓ | **9.82** | **4.11** |
| 2 | 2 | 10 | 4 | ✗ | 3.98 | 3.66 |
| 2 | 2 | 100 | 4 | ✓ | **8.52** | **4.03** |
| 2 | 2 | 100 | 4 | ✗ | -34.31 | 2.91 |
| 10 | 1 | 1 | 1 | ✓ | **10.53** | 3.65 |
| 10 | 1 | 1 | 1 | ✗ | 8.16 | **4.00** |
| 10 | 1 | 4 | 1 | ✓ | **10.72** | 3.62 |
| 10 | 1 | 4 | 1 | ✗ | 5.99 | **3.74** |
| 10 | 1 | 4 | 2 | ✓ | **10.71** | 3.73 |
| 10 | 1 | 4 | 2 | ✗ | 7.03 | **3.88** |
| 10 | 2 | 100 | 2 | ✓ | **9.23** | **4.07** |
| 10 | 2 | 100 | 2 | ✗ | -33.78 | 2.85 |
| 10 | 2 | 100 | 4 | ✓ | **9.22** | **4.09** |
| 10 | 2 | 100 | 4 | ✗ | -16.39 | 2.95 |

development of HD music streaming and video streaming services. Compression methods are used to reduce the storage requirements and network bandwidth used to serve this content. Furthermore, the growing adoption of wearable devices contributes to making efficient compression an increasingly important problem.

Different audio codecs, including Opus and EVS, have been developed and widely adopted over the past years. Those codecs support audio coding at low latency with high audio quality at low to medium bitrates (in the range of 12 to 24 kbps) but the audio quality deteriorates at very low bitrates (eg. 3 kbps) on non-speech audio. Addressing very low bitrate compression with high fidelity remains an essential challenge to solve as very low bitrate codecs enable communication and improve experiences such as videoconferencing or streaming content even with a poor internet connection and therefore allows the internet services to become more inclusive. While further work needs to be done, we hope that sharing the result of this work to the broader community can further contribute to this direction.

