# OpenReview forum: "High Fidelity Neural Audio Compression"
_TMLR — Accepted by TMLR_

### Review · Reviewer_k3bm · 2023-04-21

**Summary Of Contributions:**

This paper introduces a real-time, high-fidelity audio codec that utilizes neural networks in a streaming encoder-decoder architecture with a quantized latent space.
The training process is simplified and accelerated by employing a single multiscale spectrogram adversary, resulting in reduced artifacts and high-quality samples.
A novel loss balancer mechanism is introduced to stabilize training. The study also explores using lightweight Transformer models to compress the representation by up to 40% while maintaining real-time processing.
The paper delves into key design choices such as training objectives, architectural changes, and perceptual loss functions.
 An extensive subjective evaluation is conducted for various bandwidths and audio domains. The approach outperforms baseline methods across all settings, for both 24 kHz monophonic and 48 kHz stereophonic audio. Code and samples are provided in the supplementary material.

**Audience:**

Yes

**Broader Impact Concerns:**

The code is open source, and already gaining traction.
This is an important milestone for neural codec.

**Claims And Evidence:**

Yes

**Requested Changes:**

I have several question for the author:
The encodec Model according to my test, is 32 times slower than torchaudio's FFT (on GPU) on CPU is 23 slower.
Do you think more extreme quantization, and more aggressive distillation can make it faster? Otherwise, this will be specialized for low-bandwidth applications only.

DiffQ is weight-quantization only, without activation quatization, have your experiemented with quantizer's with activation quantization?
From the appendix, it is not clear what bit you quantized to.

Appendix typo near: For this set of experiments we use the DiffQ quantizer **other** RVQ.

Issue: 1% artifact if computed from cuda VS. CPU: using 24kHz model (8, 504) latent:
diff:<class 'torch.Tensor'>,tensor([[    0,     0,     0,     0,     0,     0,     0,     0,     0,     0,
             0,     0,     0,     0,     0,     0,     0,     0,     0,     0,
             0,     0,     0,     0,     0,     0,     0,     0,     0,     0,
             0,     0,     0,     0,     0,     0,     0,     0,     0,     0,
             0,     0,     0,     0,     0,     0,     0,     0,     0,     0,
             0,     0,     0,     0,     0,     0,     0,     0,     0,     0,
             0,     0,     0,     0,     0,     0,     0,     0,     0,     0,
             0,     0,     0,     0,     0,     0,     0,     0,     0,     0,
             0,     0,     0,     0,     0,     0,     0,     0,     0,     0,
             0,     0,     0,     0,     0,     0,     0,     0,     0,     0,
             0,     0,     0,     0,     0,     0,     0,     0,     0,     0,
             0,     0,     0,     0,     0,     0, -1305,     0,     0,     0,
             0,     0,     0,     0,     0,     0,     0,     0,     0,     0,
             0,     0,     0,     0,     0,     0,     0,     0,     0,     0,
             0,     0,     0,     0,     0,     0,     0,     0,     0,     0,
             0,     0,     0,     0,     0,     0,     0,     0,     0,     0,
             0,     0,     0,     0,     0,     0,     0,     0,     0,     0,
             0,     0,     0,     0,     0,     0,     0,     0,     0,     0,
             0,     0,     0,     0,     0,     0,     0,     0,     0,     0,
             0,     0,     0,     0,     0,     0,     0,     0,     0,     0,
             0,     0,     0,     0,     0,     0,     0,     0,     0,     0,
             0,     0,     0,     0,     0,     0,     0,     0,     0,     0,
             0,     0,     0,     0,     0,     0,     0,     0,     0,     0,
             0,     0,     0,     0,     0,     0,     0,     0,     0,     0,
             0,     0,     0,     0,     0,     0,     0,     0,     0,     0,
             0,  -457,     0,     0,     0,     0,     0,     0,     0,     0,
             0,     0,     0,     0,     0,     0,   651,     0,     0,     0,
             0,     0,     0,     0,     0,     0,     0,     0,     0,     0,
             0,     0,     0,     0,     0,     0,     0,     0,     0,     0,
             0,     0,     0,     0,     0,  -233,     0,     0,     0,     0,
             0,     0,     0,     0,     0,     0,     0,     0,     0,     0,
             0,     0,     0,     0,     0,     0,     0,     0,     0,     0,
             0,     0,     0,     0,     0,     0,     0,     0,     0,     0,
             0,     0,     0,     0,     0,     0,     0,     0,     0,     0,
             0,     0,     0,     0,     0,     0,     0,     0,     0,     0,
             0,     0,     0,     0,     0,     0,     0,     0,     0,     0,
             0,     0,     0,     0,     0,     0,     0,     0,     0,     0,
             0,     0,     0,     0,     0,     0,     0,     0,     0,     0,
             0,     0,     0,     0,     0,     0,     0,     0,     0,     0,
             0,     0,     0,     0,     0,     0,     0,     0,     0,     0,
             0,     0,     0,     0,     0,     0,     0,     0,     0,     0,
             0,     0,     0,     0,     0,     0,     0,     0,     0,     0,
             0,     0,     0,     0,     0,     0,     0,     0,     0,     0,
             0,     0,     0,     0,     0,     0,     0,     0,     0,     0,
             0,     0,     0,  2105,     0,     0,     0,     0,     0,     0,
             0,     0,     0,     0,     0,     0,     0, -1495,     0,     0,
             0,     0,     0,     0,     0,     0,     0,     0,     0,     0,
             0,     0,     0,     0,     0,     0,     0,     0,     0,     0,
             0,     0,     0,     0,     0,     0,     0,     0,     0,     0,
             0,     0,     0,     0,     0,     0,     0,     0,     0,     0,
             0,     0,     0,     0]])
the difference is huge in code book.

**Strengths And Weaknesses:**

This is an extremely useful and valuable piece of work!
System and architecture-wise, it's not surprising to someone who are familiar with audio SOTA, but it is exciting to see the idea materialize!
The loss-balancer and all these datasets' combination is a great engineering success!
I played with the code, and it works like a charm!
I am convinced!
One thing I suggest for future is to compare with Transformer-based VAE architecture such as AudioMAE (Huang et al. 2022), it would greatly benefit the community to understand which method would be preferred.
My guess is that MAE based method would need a few layers of Conv1d to be equally efficient, but might be better at retaining the global audio structure?

---

> ### Author Response · Authors · 2023-07-05
> **Author's response**
>
> **Regarding inference time:** Although we did not properly evaluate it, we believe that indeed model quantization and distillation will make model inference faster. However, we kindly disagree that in the current setup, the model will be specialized for low-bandwidth applications only, it can also be used for full-band applications, depending on the use case and the computing at the edge devices. Additionally, the fact that FFT is faster makes sense, it is a simpler operation that does not allow high-quality generations at extreme compression rates such as evaluated in this paper.
>
>
> **Regarding DiffQ:** We additionally experimented with DiffQ and Gumbel softmax to quantize the audio representation obtained from the encoder, basically replacing the RVQ layer. This acts as the audio compression layer. Hence, activation quantization is not relevant in our case.
>
> **Regarding cuda vs. CPU:** Thanks for reporting this interesting phenomenon. We were able to reproduce this with the following observations:
> - The output of the encoder will already suffer from small differences, i.e. 1e-6 relative error for the timesteps that lead to different quantization outcomes.
> - This small delta can lead to being assigned to different clusters, especially for the last few layers of RVQ which encodes increasingly small residuals.
> - The differences post quantizations are still quite small: 6% relative error for the positions where this happens, but given this is a rare event (in our tests, happening about 1/1000), the overall change in the latent space before the decoder is less than 1e-4 relative error, and in the waveform domain less than 1% relative error.

---

### Review · Reviewer_gWec · 2023-05-23

**Summary Of Contributions:**

This paper proposes an autoencoder for audio compression. The method uses discretization in the latent space, and a discriminator loss with the motivation of improving the reconstructed audio quality. The experimental results indicate that with respect to subjective scores and objective scores the method outperforms the other methods that the authors compare with. The method has streamable and non-streamable versions, and it seems the streamable version does not lose much performance over the non-streamable one.

**Audience:**

Yes

**Claims And Evidence:**

Yes

**Requested Changes:**

-As I described in section above, it would be nicer to make the reader's life a bit easier by making the paper a bit more self-contained rather than assuming knowledge on previous papers. Specifically, adding a bit more background on residual vector quantization and multiscale discriminators might help.

-It would also help to create a companion website where you provide examples from the baselines and your method.

**Strengths And Weaknesses:**

Strengths:

-The experimental results seem to suggest that the proposed method is able to outperform the other methods that the authors compare against. I am not an expert in the audio compression domain, so I am not sure if the comparisons are exhaustive.

Weaknesses:

-The motivation for the different components of the proposed method seems a bit arbitrary. (For instance the loss components) However this is okay as the experimental results seem to be good..

-It's a bit difficult to read the paper as the authors seem to assume prior knowledge of several previous papers. This includes residual vector quantization, and multiscale discriminators.

---

> ### Author Response · Authors · 2023-07-05
> **Author's response**
>
> **Regarding making the paper a bit more self-contained:** Thanks for the suggestion. We included new paragraphs explaining RVQ and the discriminators in more detail.
>
> **Regarding adding a sample page:** We included a website where we compare the proposed method against the evaluated baselines. It is on the supplemental material. Upon acceptance, we will publish the website.

---

### Review · Reviewer_6m85 · 2023-06-14

**Summary Of Contributions:**

This paper introduced a novel audio codec to achieve real-time, high-fidelity audio compression. The technical contribution of this paper lies in: 1) They simplify the training of GAN by a single multiscale spectrogram adversary (MS-STFT). 2) They introduce a novel loss balancer mechanism to stabilize the training. 3) They propose a lightweight transformer-based LM and entropy encoding mechanism to further compress the obtained representation. Overall, this paper presents a remarkable advancement in audio compression, showcasing innovative techniques that simplify GAN training, introduce a loss balancer mechanism, and leverage lightweight transformer-based LM and entropy encoding. The extensive evaluation further solidifies the effectiveness of the proposed codec, making it a significant contribution to the field of audio compression.

**Audience:**

Yes

**Broader Impact Concerns:**

No.

**Claims And Evidence:**

Yes

**Requested Changes:**

1)	Please refer to weaknesses 1 and 2, the authors should provide more explanation of LM and entropy encoding.
2)	Please refer to weakness 3. The authors should include more explanation and verification of the normalization operation.
3)	Please refer to weakness 4. The codebook replacement should be more detailed to avoid confusion.


**Strengths And Weaknesses:**

Strength:

1)	The simplification of adversarial loss with extensive experimental demonstration is convincing and effective.
2)	The loss balancer is effective to stabilize the codec training. It can be extended to other audio compression works too.
3)	The lightweight LM with entropy encoding is novel. With the help of such a design, the RTF can be reduced.
4)	The paper is well written, and the code is released.
5)	The evaluation and ablation study conducted for a range of audio domains demonstrate the superiority of the proposed approach over baseline methods across all evaluated settings.

Weakness:

1)	The insight and methodology of the proposed language model are not clear and straightforward. In this paper, the LM seems to predict latent representation at time t given 1 to t-1 times discrete representations. Then it will predict N_q discrete tokens in parallel at time t without using any audio information of time t. But without time t’s audio, will the results degrade? Will the predicted time t’s discrete representation be used to predict time t+1’s representation?
2)	The entropy encoding is also confusing. The paper uses entropy encoding to compress the probability given by LM. But the codec only uses the discrete tokens. Why the probability of LM instead of the discrete tokens should be compressed?
3)	In Sec. 3.1 Non-streamable paragraph, EnCodec will first normalize the chunk before feeding it to the model and apply an inverse operation on the output, which is not a familiar operation in other codec works. Why is it important? More explanation should be included.
4)	In Sec. 3.2, when training Encodec, the entries in the codebook that are not used are replaced with a candidate sampled from the current batch. Will this replacement happen in every iteration? It seems too frequent to update the code in every batch iteration.

---

> ### Author Response · Authors · 2023-07-05
> **Author's response**
>
> **Regarding LM prediction:** The LM’s role is to model the distribution of tokens at time t based on previous tokens (1 to t-1). During training time, the LM predicts exactly that. During inference time, we leverage this distribution to reach better compression rates of the discrete tokens than assuming a uniform distribution. Notice, we do not use the LM predictions as input to the decoder, hence no reason to suffer from degradation in results, only better compression rates.
>
> **Regarding compression LM probabilities:** We believe there is a misunderstanding. We do not compress the LM probabilities. Instead of assuming uniform distribution of the tokens, we leverage the LM probabilities over the tokens to reach better compression rates of the tokens. We clarified that in the paper.
>
> **Regarding normalization:** in early experiments, we find input normalization and output denormalization to provide superior results and make model optimization easier. This approach was also found useful in [1]. We included a paragraph about that in the paper.
>
> **Regarding replacement of unused tokens:** We follow the method proposed by [2]. We track the exponential moving average of the assignments to each vector and replace the vectors for which the average cluster size falls below 2.
>
>
> [1] Defossez, Alexandre, Gabriel Synnaeve, and Yossi Adi. "Real time speech enhancement in the waveform domain." Interspeech (2020).
>
> [2] Prafulla Dhariwal, Heewoo Jun, Christine Payne, Jong Wook Kim, Alec Radford, and Ilya Sutskever. Jukebox: A generative model for music, 2020

---

### Author Response · Authors · 2023-07-05
**Author's response**

We would like to thank the reviewers for their helpful comments and suggestions! We addressed the main issues, fixed typos, and updated the paper. A dedicated response to each of the reviewers can be found below.

---

### Decision · Action_Editors · 2023-07-25

**Recommendation:** Accept as is

**Comment:**

See the comments above.

**Audience:**

It would be interesting to a broad range of audiences in the audio community.

**Claims And Evidence:**

All three reviewers acknowledge that this is good work for high-fidelity audio compression, which has already received wide attention and adoption in the audio community. The authors respond to the reviewers' questions actively and resolve the questions and concerns. Overall, it is a strong paper.

---

> ### Author Response · Authors · 2023-08-05
> **Response to Action Editor**
>
> Thanks for the valuable feedback and for accepting our manuscript!